

# Description of three new species of Geometridae (Lepidoptera) using species delimitation in an integrative taxonomy approach for a cryptic species complex

Simeão S. Moraes[1], Ygor Montebello[1], Mariana A. Stanton[2], Lydia Fumiko Yamaguchi[2], Massuo J. Kato[2] and André V.L. Freitas[1]

[1] Departamento de Biologia Animal and Museu da Biodiversidade, Universidade Estadual de Campinas, Campinas, São Paulo, Brazil

[2] Laboratório de Química de Produtos Naturais, Instituto de Química, Universidade de São Paulo, São Paulo, São Paulo, Brazil

## ABSTRACT

The genus *Eois* Hübner (Geometridae: Larentiinae) comprises 254 valid species, 217 of which were described from the Neotropics and 31 of those having their type locality in Brazil. Since this species rich genus has never been revised, and may potentially include many cryptic undescribed species, *Eois* embodies a problematic taxonomic scenario. The actual diversity of *Eois* is greatly underestimated and the Brazilian fauna is poorly known, both because of inadequate sampling and because of the potential existence of cryptic species "hidden" within some nominal taxa. In this study we investigated the diversity within a cryptic species complexes associated to the *E. pallidicosta* and *E. odatis* clades. We describe three new species *Eois oya* Moraes & Montebello **sp. nov.**, *Eois ewa* Moraes & Stanton **sp. nov.**, and *Eois oxum* Moraes & Freitas **sp. nov.,** in an integrative taxonomy approach, using morphology, host plant use and species delimitation tools.

## INTRODUCTION

Geometridae is a megadiverse family with over 24,000 species, being the most species rich lineage in the superfamily Geometroidea and the second most species-rich family among Macroheterocera lineages (*Scoble, 1999*; *Sihvonen et al., 2011*; *Mitter, Davis & Cummings, 2017*). The family Geometridae represents a challenge for researchers because of the taxonomic uncertainties around some species-rich genera, for which taxonomic reviews are lacking and several new taxa await formal descriptions (*Strutzenberger et al., 2011*; *Strutzenberger, Brehm & Fiedler, 2012*). Despite the existence of a worldwide catalog (*Scoble, 1999*), only a small subset of Geometridae genera has been revised (*Hulst, 1896*; *Wehrli, 1939*; *Rindge, 1990*, *Pitkin, 2002*). Most of these taxonomic studies are restricted to specific geographic areas, leaving the Neotropics as one of the least studied biogeographic regions for the Geometridae fauna.

Corresponding author
Simeão S. Moraes,
simeao_moraes@yahoo.com.br

*Eois* Hübner belongs to the subfamily Larentiinae and is one of the most species-rich genus, with 254 described species (*Brehm et al., 2011*), 217 of which were described for the Neotropical region and 31 primary types for Brazil. Species of *Eois* are small, reaching up to 2 cm in wingspan, and most species present a wing pattern consisting of a yellowish or brown background patterned with reddish or rusty symmetrical bands on both wings. Based on a study with some species from Borneo (*Holloway, 1997*), the male genitalia lack the uncus and labides, while female genitalia present a robust bursa with multiseriate signa. Recently, *Brehm et al. (2011)* showed that characters from valvae and vesica on the male genitalia might be phylogenetically informative and also useful in delimiting species.

Notwithstanding *Eois* being recognized as an important component of the diversity of neotropical moths (*Brehm et al., 2011*), there is still a large gap in the knowledge and representativeness of the Brazilian fauna. Single-locus species delimitation methods were recognized as a useful approach when working with montane *Eois* species (*Strutzenberger et al., 2011*) and recently this diversity was also increased for lowland species (*Moraes et al., 2021*). However, despite the use of molecular methods for species-delimitation having proven to be extremely valuable in highlighting cryptic diversity, fewer than 30% of the studies on species delimitation made taxonomic recommendations and only 25% described new species (*Carstens et al., 2013*).

In the present study we contribute to the knowledge of *Eois* diversity using an integrative taxonomic approach on cryptic species complexes, and describe three new species of *Eois*. Accordingly, we accessed the molecular diversity and used the Automatic Barcode Gap Discovery method (ABGD) for species delimitation (*Puillandre et al., 2012*). To test the validity of the molecular taxonomic units (MOTUs) we studied morphological characters for wing pattern and genitalia, as well as host plant use by larvae. We expect that these practices will improve the number of new taxa described in cryptic complexes, increasing the knowledge on the real diversity of *Eois* from Brazil.

## MATERIALS AND METHODS

### Sampling

Immature stages and imagos of *Eois* were obtained from six localities: (i) Serra do Japi Biological Reserve (23° 14′ S 46° 58′ W), Jundiaí, São Paulo, Brazil; (ii) Boraceia Biological Station (23° 39′ S 45° 54′ W), Salesópolis, São Paulo, Brazil; (iii) Intervales State Park (24° 16′ S 48° 24′ W), Ribeirão Grande, São Paulo, Brazil, (iv) Itatiaia National Park (22° 27′ S 44° 37′ W), Itatiaia, Rio de Janeiro, Brazil, (v) Serra dos Orgãos National Park (22° 27′ S 42° 59′ W), Teresópolis, Rio de Janeiro, Brazil, and (vi) Adolpho Ducke Forest Reserve (2° 57′ S 59° 55′ W), Manaus, Amazonas, Brazil.

Imagos were collected during the night using a light source consisting of a a 500 W mix bulb on a square white sheet (2 m side). Immatures were collected by searching plants of the genus *Piper* L. and *Peperomia* Ruiz & Pav.(Piperaceae), known larval host plants of several species of *Eois*, and were reared in the laboratory. Samples of the host plant, including leaves, inflorescence and fruits, were also collected for further identification. Permits for field trips were issued by Instituto Chico Mendes de Conservação da

Biodiversidade (ICMBio, permit nos. 10362-1, 15780-10 and 22205-6). All new described species are registered in the SISGEN (A4ED092).

## Rearing methods

Larvae were reared in individual 300 mL clear plastic vials with lids, and provided with leaves of the same *Piper* and *Peperomia* plants on which they were collected (*Moraes, Otero & Freitas, 2017*). Pupae were transferred to individual 50 mL clear plastic vials with lids and moist cotton wool until emergence. Larvae and pupae were maintained under constant temperature (25 °C) and 12 h light: 12 h dark cycle. After emergence, imagos were sacrificed for DNA extraction and genitalia dissection. Samples of the host plants were collected and compared to previously identified samples from the same locality made at the Laboratório de Química de Produtos Naturais, at the University of São Paulo, and compared to the species description in the Brazilian Flora 2020 project (*Guimarães, Queiroz & Medeiros, 2020*).

## DNA extraction and PCR conditions

Three legs were removed from each specimen shortly after collection and before spreading. Sampled legs were preserved dry and stored in 1.5 mL tubes at −20 °C. Total genomic DNA was extracted with DNeasy Blood & Tissue Kit (Qiagen, Venlo, Netherlands), according to the manufacturer's protocol with final elution in 100 µl elution buffer. The 5′ end (barcode region) of the mitochondrial gene cytochrome oxidase subunit I (COI, 650 bp) was amplified using the primers HCO and LCO (*Folmer et al., 1994*) containing the T3 and T7 promotor universal tails, respectively (*Wahlberg & Wheat, 2008*). Polymerase chain reactions (PCRs) were performed with 13 µl total volume containing 1–2 µl of extracted DNA, 3.2–4.2 µl of $H_2O$ milli-Q, 6.5 µl of 2x MyTaq HS red mix (Bioline Co., London, UK), and 0.65 µl of each primer (10 mM). PCR products were amplified as follows: 96 °C for 7 min, followed by 40 cycles of 96 °C for 30 s, 50 °C for 30 s and 72 °C for 90 s, and a final extension period of 72 °C for 10 min.

Amplicons were purified adding a mix of 1.3 µl of FastAP Thermosensitive Alkaline Phosphatase (Thermofisher Scientific, Waltham, MA, USA) and 0.7 µl of Exonuclease 1 (Thermofisher Scientific, Waltham, MA, USA) to 10 µl of PCR products. Purified products were sent for Sanger sequencing.

## Alignment, tree inference and species delimitation

The genetic dataset consisted of 160 newly sequenced individuals of 18 putative species combined with 36 sequences obtained from Genbank (Table 1). Sequences were aligned using MAFFT (*Katoh et al., 2002*) implemented in Geneious v.11.0.2 (*Kearse et al., 2012*). The alignments were carefully checked by eye, taking into consideration the reading frame relative to the reference sequence. The Maximum Likelihood analyses were conducted using RAxML-HPC2 V.8.2.10 (*Stamatakis, 2014*) on the webserver CIPRES Science Gateway (*Miller, Pfeiffer & Schwartz, 2010*). Support for nodes was evaluated with 1000 ultrafast bootstrap (UFBoot2) approximations (*Hoang et al., 2018*), UFBoot2 values ≥95 indicate well-supported clades.

**Table 1 List of specimens vouchered, the associated clade, geographical localities, host plant and herbarium voucher for host plants used in this study.**

| GenBank Voucher | Clade | Locality | Host Plant | Herbarium Voucher |
|---|---|---|---|---|
| LEPSM818 | | | | |
| LEPSM609 | | | | |
| K2422-1 | olivacea | Brazil, MS, Aquidauana | *Piper tuberculatum* Jacq. | K2422 |
| K2422-1a | olivacea | Brazil, MS, Aquidauana | *Piper tuberculatum* Jacq. | K2422 |
| K2426-2 | olivacea | Brazil, MS, Aquidauana | *Piper tuberculatum* Jacq. | K2422 |
| K2426-3 | olivacea | Brazil, MS, Aquidauana | *Piper tuberculatum* Jacq. | K2422 |
| M097-1 | russearia | Brazil, SP, Mogi-Guaçu, Reserva Biológica de Mogi-Guaçu | *Piper arboreum* Aubl. | K1953* |
| Mogi#906 | russearia | Brazil, SP, Mogi-Guaçu, Reserva Biológica de Mogi-Guaçu | *Piper arboreum* Aubl. | K1953* |
| M081-1 | russearia | Brazil, SP, Mogi-Guaçu, Reserva Biológica de Mogi-Guaçu | *Piper crassinervium* Kunth. | K1954* |
| Mogi#802 | russearia | Brazil, SP, Mogi-Guaçu, Reserva Biológica de Mogi-Guaçu | *Piper crassinervium* Kunth. | K1954* |
| M076-1 | russearia | Brazil, SP, Mogi-Guaçu, Reserva Biológica de Mogi-Guaçu | *Piper crassinervium* Kunth. | K1954* |
| M081-2 | russearia | Brazil, SP, Mogi-Guaçu, Reserva Biológica de Mogi-Guaçu | *Piper crassinervium* Kunth. | K1954* |
| Mogi#818 | russearia | Brazil, SP, Mogi-Guaçu, Reserva Biológica de Mogi-Guaçu | *Piper crassinervium* Kunth. | K1954* |
| K2411-2 | russearia | Brazil, MS, Aquidauana | *Piper sp 2* | K2417 |
| K2417-12 | russearia | Brazil, MS, Aquidauana | *Piper sp 2* | K2417 |
| K2417-7 | russearia | Brazil, MS, Aquidauana | *Piper sp 2* | K2417 |
| K2417-27 | russearia | Brazil, MS, Aquidauana | *Piper sp 2* | K2417 |
| K2416-16 | russearia | Brazil, MS, Aquidauana | *Piper sp 2* | K2417 |
| LEPSM906 | adimaria | | | |
| LEPSM907 | adimaria | | | |
| K2206-1 | odatis | Brazil, SP, Jundiaí, Reserva Biológica da Serra do Japi | *Piper lindbergii* C.DC | K2493 |
| K2325-15 | odatis | Brazil, SP, Jundiaí, Reserva Biológica da Serra do Japi | *Piper lindbergii* C.DC | K2493 |
| K2325-9 | odatis | Brazil, SP, Jundiaí, Reserva Biológica da Serra do Japi | *Piper lindbergii* C.DC | K2493 |
| K2325-16 | odatis | Brazil, SP, Jundiaí, Reserva Biológica da Serra do Japi | *Piper lindbergii* C.DC | K2493 |
| K2325-10 | odatis | Brazil, SP, Jundiaí, Reserva Biológica da Serra do Japi | *Piper lindbergii* C.DC | K2493 |
| K2206-23 | odatis | Brazil, SP, Jundiaí, Reserva Biológica da Serra do Japi | *Piper lindbergii* C.DC | K2493 |
| M768-1 | odatis | Brazil, SP, Jundiaí, Reserva Biológica da Serra do Japi | *Piper lindbergii* C.DC | M1033 |
| M528-4 | odatis | Brazil, RJ, Itatiaia, Parque Nacional do Itatiaia | *Piper amplum* Kunth. | M1033 |
| M528-1 | odatis | Brazil, RJ, Itatiaia, Parque Nacional do Itatiaia | *Piper amplum* Kunth. | M1033 |
| LEPSM909 | odatis | | | |
| K2325-2 | odatis | Brazil, SP, Jundiaí, Reserva Biológica da Serra do Japi | *Piper lindbergii* C.DC | K2493 |
| K2452-1 | odatis | Brazil, RJ, Itatiaia, Parque Nacional do Itatiaia | *Peperomia hispidula* (Sw.) A. Dietr. | K1612* |
| K2452-4 | odatis | Brazil, RJ, Itatiaia, Parque Nacional do Itatiaia | *Peperomia hispidula* (Sw.) A. Dietr. | K1612* |

| GenBank Voucher | Clade | Locality | Host Plant | Herbarium Voucher |
|---|---|---|---|---|
| K2452-5 | odatis | Brazil, RJ, Itatiaia, Parque Nacional do Itatiaia | *Peperomia hispidula* (Sw.) A. Dietr. | K1612* |
| K2452-2 | odatis | Brazil, RJ, Itatiaia, Parque Nacional do Itatiaia | *Peperomia hispidula* (Sw.) A. Dietr. | K1612* |
| M624-1 | odatis | Brazil, RJ, Itatiaia, Parque Nacional do Itatiaia | *Piper malacophyllum* C. Presl | M1038 |
| M289-1 | odatis | Brazil, SP, São Paulo, Parque Estadual do Jaraguá | *Piper malacophyllum* C. Presl | K2294* |
| #107 | odatis | | | |
| M621-1 | odatis | Brazil, RJ, Itatiaia, Parque Nacional do Itatiaia | *Piper malacophyllum* C. Presl | M1038 |
| M436-1 | odatis | Brazil, SP, São Paulo, Parque Estadual do Jaraguá | *Piper malacophyllum* C. Presl | K2294* |
| K2159-7 | odatis | Brazil, RJ, Itatiaia, Parque Nacional do Itatiaia | *Piper malacophyllum* C. Presl | K2159 |
| M576-1 | odatis | Brazil, RJ, Itatiaia, Parque Nacional do Itatiaia | *Piper malacophyllum* C. Presl | M1038 |
| M657-1 | odatis | Brazil, SP, São Paulo, Parque Estadual do Jaraguá | *Piper gaudichaudianum* Kunth. | K2311* |
| M435-1 | odatis | Brazil, SP, São Paulo, Parque Estadual do Jaraguá | *Piper malacophyllum* C. Presl | K2294* |
| M622-1 | odatis | Brazil, RJ, Itatiaia, Parque Nacional do Itatiaia | *Piper malacophyllum* C. Presl | M1038 |
| M388-1 | odatis | Brazil, SP, São Paulo, Parque Estadual do Jaraguá | *Piper gaudichaudianum* Kunth. | K2311* |
| K2157-2 | odatis | Brazil, RJ, Itatiaia, Parque Nacional do Itatiaia | *Piper malacophyllum* C. Presl | K2165 |
| PPU721 | odatis | Brazil, SP, Jundiaí, Reserva Biológica da Serra do Japi | *Piper malacophyllum* C. Presl | K2306 |
| M761-1 | odatis | Brazil, SP, Jundiaí, Reserva Biológica da Serra do Japi | *Piper gaudichaudianum* Kunth. | K2198* |
| M754-1 | odatis | Brazil, SP, Jundiaí, Reserva Biológica da Serra do Japi | *Piper gaudichaudianum* Kunth. | K2198* |
| PPU660 | odatis | Brazil, SP, Jundiaí, Reserva Biológica da Serra do Japi | *Piper malacophyllum* C. Presl | K2306 |
| M358-1 | odatis | Brazil, SP, São Paulo, Parque Estadual do Jaraguá | *Piper cubataonum* C. DC | K1951* |
| M775-1 | odatis | Brazil, SP, Jundiaí, Reserva Biológica da Serra do Japi | *Piper gaudichaudianum* Kunth. | K2198* |
| M678-1 | odatis | Brazil, SP, São Paulo, Parque Estadual do Jaraguá | *Piper malacophyllum* C. Presl | K2294* |
| M677-1 | odatis | Brazil, SP, São Paulo, Parque Estadual do Jaraguá | *Piper gaudichaudianum* Kunth. | K2311* |
| K2165-3 | odatis | Brazil, RJ, Itatiaia, Parque Nacional do Itatiaia | *Piper malacophyllum* C. Presl | K2165 |
| M514-1 | odatis | Brazil, RJ, Itatiaia, Parque Nacional do Itatiaia | *Piper crassinervium* Kunth. | K1954* |
| K2423-5 | odatis | Brazil, MS, Corumbá | *Piper amalago* L. | K2421 |
| K2423-4 | odatis | Brazil, MS, Corumbá | *Piper amalago* L. | K2421 |
| K2423-8 | odatis | Brazil, MS, Corumbá | *Piper amalago* L. | K2421 |
| K2423-3 | odatis | Brazil, MS, Corumbá | *Piper amalago* L. | K2421 |
| K2212-11 | odatis | Brazil, SP, Jundiaí, Reserva Biológica da Serra do Japi | *Piper malacophyllum* C. Presl | K2212 |
| M616-1 | odatis | Brazil, RJ, Itatiaia, Parque Nacional do Itatiaia | *Piper chimonanthifolium* Kunth. | K1960* |
| M399-1 | odatis | Brazil, SP, São Paulo, Parque Estadual do Jaraguá | *Piper malacophyllum* C. Presl | K2294* |
| #359 | odatis | | | |
| M731-1 | odatis | Brazil, SP, Jundiaí, Reserva Biológica da Serra do Japi | *Piper chimonanthifolium* Kunth. | K1960* |
| K2454-38 | odatis | Brazil, SP, Campos do Jordão | *Piper gaudichaudianum* Kunth. | K2446 |
| K2454-31 | odatis | Brazil, SP, Campos do Jordão | *Piper gaudichaudianum* Kunth. | K2446 |
| K2446-7 | odatis | Brazil, SP, Campos do Jordão | *Piper gaudichaudianum* Kunth. | K2446 |
| K2454-47 | odatis | Brazil, SP, Campos do Jordão | *Piper gaudichaudianum* Kunth. | K2446 |
| K2453-47 | odatis | Brazil, SP, Campos do Jordão | *Piper gaudichaudianum* Kunth. | K2446 |
| K2165-5 | odatis | Brazil, RJ, Itatiaia, Parque Nacional do Itatiaia | *Piper malacophyllum* C. Presl | K2165 |

(Continued)

| GenBank Voucher | Clade | Locality | Host Plant | Herbarium Voucher |
|---|---|---|---|---|
| | | **Table 1 (continued)** | | |
| M611-1 | odatis | Brazil, RJ, Itatiaia, Parque Nacional do Itatiaia | *Piper chimonanthifolium* Kunth. | K1960* |
| #845 | odatis | Brazil, SP, Mogi-Guaçu, Reserva Biológica de Mogi-Guaçu | *Piper gaudichaudianum* Kunth. | |
| #855 | odatis | Brazil, SP, Mogi-Guaçu, Reserva Biológica de Mogi-Guaçu | *Piper gaudichaudianum* Kunth. | |
| #841 | odatis | Brazil, SP, Mogi-Guaçu, Reserva Biológica de Mogi-Guaçu | *Piper gaudichaudianum* Kunth. | |
| PPU651 | odatis | Brazil, SP, Jundiaí, Reserva Biológica da Serra do Japi | *Piper hillianum* C. DC. | K1920* |
| M844-2 | odatis | Brazil, SP, Capão Bonito, Parque Estadual Intervales | *Piper gaudichaudianum* Kunth. | M1034 |
| M617-1 | odatis | Brazil, RJ, Itatiaia, Parque Nacional do Itatiaia | *Piper chimonanthifolium* Kunth. | K1960* |
| M735-1 | odatis | Brazil, SP, Jundiaí, Reserva Biológica da Serra do Japi | *Piper chimonanthifolium* Kunth. | K2495 |
| M738-1 | odatis | Brazil, SP, Jundiaí, Reserva Biológica da Serra do Japi | *Piper chimonanthifolium* Kunth. | K2495 |
| M618-1 | odatis | Brazil, RJ, Itatiaia, Parque Nacional do Itatiaia | *Piper chimonanthifolium* Kunth. | K1960* |
| M573-1-4 | odatis | Brazil, RJ, Itatiaia, Parque Nacional do Itatiaia | *Piper chimonanthifolium* Kunth. | K1960* |
| M573-1-3 | odatis | Brazil, RJ, Itatiaia, Parque Nacional do Itatiaia | *Piper chimonanthifolium* Kunth. | K1960* |
| K2453-4-1 | odatis | Brazil, SP, Campos do Jordão | *Piper gaudichaudianum* Kunth. | K2446 |
| M573-1-5 | odatis | Brazil, RJ, Itatiaia, Parque Nacional do Itatiaia | *Piper chimonanthifolium* Kunth. | K1960* |
| M163-1-1 | odatis | Brazil, RJ, Itatiaia, Parque Nacional do Itatiaia | *Piper chimonanthifolium* Kunth. | K1960* |
| K2453-4b | odatis | Brazil, SP, Campos do Jordão | *Piper gaudichaudianum* Kunth. | K2446 |
| K2453-4-2 | odatis | Brazil, SP, Campos do Jordão | *Piper gaudichaudianum* Kunth. | K2446 |
| K2453-4 | odatis | Brazil, SP, Campos do Jordão | *Piper gaudichaudianum* Kunth. | K2446 |
| M573-1-2 | odatis | Brazil, RJ, Itatiaia, Parque Nacional do Itatiaia | *Piper chimonanthifolium* Kunth. | K1960* |
| K2453-4-3 | odatis | Brazil, SP, Campos do Jordão | *Piper gaudichaudianum* Kunth. | K2446 |
| LEPSM920 | | | | |
| #25 | | | | |
| M022-1 | veniliata | Brazil, SP, São Paulo, Parque Estadual do Jaraguá | *Piper gaudichaudianum* Kunth. | K2494* |
| M286-1 | veniliata | Brazil, SP, São Paulo, Parque Estadual do Jaraguá | *Piper gaudichaudianum* Kunth. | K2494* |
| K2326-1 | veniliata | Brazil, SP, Jundiaí, Reserva Biológica da Serra do Japi | *Piper gaudichaudianum* Kunth. | K2494* |
| K2322-1 | hyperythraria | Brazil, SP, Jundiaí, Reserva Biológica da Serra do Japi | *Piper arboreum* Aubl. | K1953* |
| M522-3 | hyperythraria | Brazil, RJ, Itatiaia, Parque Nacional do Itatiaia | *Piper cernuum* Vell. | K1925* |
| K2322-5 | hyperythraria | Brazil, SP, Jundiaí, Reserva Biológica da Serra do Japi | *Piper arboreum* Aubl. | K1953* |
| LEPSM1094 | hyperythraria | | | |
| K2322-17 | hyperythraria | Brazil, SP, Jundiaí, Reserva Biológica da Serra do Japi | *Piper arboreum* Aubl. | K1953* |
| #798 | hyperythraria | Brazil, SP, Mogi-Guaçu, Reserva Biológica de Mogi-Guaçu | *Piper arboreum* Aubl. | K1953* |
| M589-3 | hyperythraria | Brazil, RJ, Itatiaia, Parque Nacional do Itatiaia | *Piper arboreum* Aubl. | K1953* |
| M644-2a | hyperythraria | Brazil, RJ, Itatiaia, Parque Nacional do Itatiaia | *Piper arboreum* Aubl. | K1953* |
| M591-1 | hyperythraria | Brazil, RJ, Itatiaia, Parque Nacional do Itatiaia | *Piper truncatum* Vell. | K1950* |
| LEPSM901 | hyperythraria | | | |
| K2366-1 | olivacea | Brazil, SP, Capão Bonito, Parque Estadual Intervales | *Piper aduncum* L. | K2387 |
| K2369-1 | olivacea | Brazil, SP, Capão Bonito, Parque Estadual Intervales | *Piper aduncum* L. | K2387 |
| K2369-3 | olivacea | Brazil, SP, Capão Bonito, Parque Estadual Intervales | *Piper aduncum* L. | K2387 |

| GenBank Voucher | Clade | Locality | Host Plant | Herbarium Voucher |
|---|---|---|---|---|
| M523-2 | olivacea | Brazil, RJ, Itatiaia, Parque Nacional do Itatiaia | *Piper cernuum* Vell. | K1925* |
| LEPSM615 | olivacea | | | |
| M527-4 | olivacea | Brazil, RJ, Itatiaia, Parque Nacional do Itatiaia | *Piper cernuum* Vell. | K1925* |
| M572-2 | olivacea | Brazil, RJ, Itatiaia, Parque Nacional do Itatiaia | *Piper cernuum* Vell. | K1925* |
| LEPSM618 | olivacea | | | |
| M516-1 | olivacea | Brazil, RJ, Itatiaia, Parque Nacional do Itatiaia | *Piper crassinervium* Kunth. | K1954* |
| K2423-2 | olivacea | Brazil, MS, Corumbá | *Piper amalago* L. | K2423 |
| M544-1 | olivacea | Brazil, RJ, Itatiaia, Parque Nacional do Itatiaia | *Piper crassinervium* Kunth. | K1954* |
| K2367-3 | olivacea | Brazil, SP, Capão Bonito, Parque Estadual Intervales | *Piper crassinervium* Kunth. | K1954* |
| K2372-10 | olivacea | Brazil, SP, Capão Bonito, Parque Estadual Intervales | *Piper crassinervium* Kunth. | K1954* |
| K2367-5 | olivacea | Brazil, SP, Capão Bonito, Parque Estadual Intervales | *Piper crassinervium* Kunth. | K1954* |
| K2322-2 | olivacea | Brazil, SP, Jundiaí, Reserva Biológica da Serra do Japi | *Piper arboreum* Aubl. | K1953* |
| K2372-6 | olivacea | Brazil, SP, Capão Bonito, Parque Estadual de Intervales | *Piper crassinervium* Kunth. | K1954* |
| K2372-8 | olivacea | Brazil, SP, Capão Bonito, Parque Estadual Intervales | *Piper crassinervium* Kunth. | K1954* |
| #862 | olivacea | Brazil, SP, Mogi-Guaçu, Reserva Biológica de Mogi-Guaçu | *Piper crassinervium* Kunth. | |
| M551-2 | olivacea | Brazil, RJ, Itatiaia, Parque Nacional do Itatiaia | *Piper crassinervium* Kunth. | K1954* |
| M548-1 | olivacea | Brazil, RJ, Itatiaia, Parque Nacional do Itatiaia | *Piper crassinervium* Kunth. | K1954* |
| M626-1 | olivacea | Brazil, RJ, Itatiaia, Parque Nacional do Itatiaia | *Piper crassinervium* Kunth. | K1954* |
| M593-1 | olivacea | Brazil, RJ, Itatiaia, Parque Nacional do Itatiaia | *Piper* cf. *tectoniifolium* Kunth. | K1958* |
| K1228-2 | olivacea | Colombia, Bogota | *Piper bogotense* C. DC. | |
| JQ424371.1 | pallidicosta | | | |
| JQ424372.1 | pallidicosta | | | |
| K1228-1 | pallidicosta | Colombia, Bogota | *Piper bogotense* C. DC. | |
| JQ179873.1 | pallidicosta | | | |
| JF859258.1 | pallidicosta | | | |
| JQ424375.1 | pallidicosta | | | |
| GQ433544.1 | pallidicosta | | | |
| KU381316.1 | pallidicosta | | | |
| JQ179915.1 | pallidicosta | | | |
| MG572804.1 | pallidicosta | | | |
| JQ179875.1 | pallidicosta | | | |
| JQ179870.1 | pallidicosta | | | |
| KU381055.1 | pallidicosta | | | |
| HQ576249.1 | pallidicosta | | | |
| MG572805.1 | pallidicosta | | | |
| JX150931.1 | pallidicosta | | | |
| KU381753.1 | pallidicosta | | | |
| JX150863.1 | pallidicosta | | | |
| GQ433565 | pallidicosta | | | |
| GQ433565.1 | pallidicosta | | | |

(Continued)

| GenBank Voucher | Clade | Locality | Host Plant | Herbarium Voucher |
|---|---|---|---|---|
| JQ179924.1 | pallidicosta | | | |
| MG572814.1 | pallidicosta | | | |
| JQ179903.1 | pallidicosta | | | |
| JQ179880.1 | pallidicosta | | | |
| GQ433564.1 | pallidicosta | | | |
| JQ424379.1 | pallidicosta | | | |
| JQ424374.1 | pallidicosta | | | |
| M157-3 | pallidicosta | Brazil, RJ, Itatiaia, Parque Nacional do Itatiaia | *Piper reitzii* Yunck. | M157 |
| LEPSM922 | pallidicosta | | | |
| M156-5 | pallidicosta | Brazil, RJ, Itatiaia, Parque Nacional do Itatiaia | *Piper reitzii* Yunck. | M156 |
| M155-12 | pallidicosta | Brazil, RJ, Itatiaia, Parque Nacional do Itatiaia | *Piper reitzii* Yunck. | M158 |
| M155-3 | pallidicosta | Brazil, RJ, Itatiaia, Parque Nacional do Itatiaia | *Piper reitzii* Yunck. | M158 |
| JQ179895.1 | pallidicosta | | | |
| JQ179923.1 | pallidicosta | | | |
| JQ179901.1 | pallidicosta | | | |
| JQ179902.1 | pallidicosta | | | |
| JQ179905.1 | pallidicosta | | | |
| JQ179906.1 | pallidicosta | | | |
| GQ433557.1 | pallidicosta | | | |
| JQ179891.1 | pallidicosta | | | |
| JQ179941.1 | pallidicosta | | | |
| M594-1 | olivacea | Brazil, RJ, Itatiaia, Parque Nacional do Itatiaia | *Piper cubataonum* C DC. | K1951* |
| M357-2 | olivacea | Brazil, SP, São Paulo, Parque Estadual do Jaraguá | *Piper cubataonum* C DC. | K1951* |
| M595-1 | olivacea | Brazil, RJ, Itatiaia, Parque Nacional do Itatiaia | *Piper cubataonum* C DC. | K1951* |
| M594-2 | olivacea | Brazil, RJ, Itatiaia, Parque Nacional do Itatiaia | *Piper cubataonum* C DC. | K1951* |
| K2420-27 | russearia | Brazil, MS, Aquidauana | *Piper sp 1* | K2420 |
| K2420-3 | russearia | Brazil, MS, Aquidauana | *Piper sp 1* | K2420 |
| K2420-21 | russearia | Brazil, MS, Aquidauana | *Piper sp 1* | K2420 |
| K2420-a | russearia | Brazil, MS, Aquidauana | *Piper sp 1* | K2420 |
| TMD2018#4 | russearia | Brazil, AM, Manaus, Reserva Florestal Adolpho Ducke | *Piper erectipilum* Yunck. | M1041 |
| TMD2018#5 | russearia | Brazil, AM, Manaus, Reserva Florestal Adolpho Ducke | *Piper erectipilum* Yunck. | M1041 |
| TMD2018#6 | russearia | Brazil, AM, Manaus, Reserva Florestal Adolpho Ducke | *Piper erectipilum* Yunck. | M1041 |
| TMD2018#9 | russearia | Brazil, AM, Manaus, Reserva Florestal Adolpho Ducke | *Piper erectipilum* Yunck. | M1041 |
| TMD2018#12 | russearia | Brazil, AM, Manaus, Reserva Florestal Adolpho Ducke | *Piper erectipilum* Yunck. | M1041 |
| TMD2018#20 | russearia | Brazil, AM, Manaus, Reserva Florestal Adolpho Ducke | *Piper erectipilum* Yunck. | M1041 |

**Note:**
Herbarium specimens that were identified by comparison to previously collected samples deposited at the University of São Paulo Herbarium (SPF) are reported with (*).

In order to study the species boundaries within our dataset, we used a species delimitation method focused on single-locus gene analysis: The Automatic Barcode Gap Discovery (ABGD, *Puillandre et al., 2012*). This method appeared to be more congruent with the *Eois* morphology compared to other methods (mPTP and bPTP)

 

(*Moraes et al., 2021*). The ABGD method seeks to quantify a range of the barcode gap that separates intra from interspecific distances, automatically clustering sequences into candidate species based on pairwise distances (*Puillandre et al., 2012*). Default settings were used for the prior range for maximum intraspecific divergence (0.001, 0.1). Results were compared using Jukes-Cantor (JC69) corrected distances and relative gap width of 1.0. ABGD analyses were performed using the graphic web version (https://bioinfo. mnhn.fr/abi/public/abgd/).

## Morphological study

The external morphology and color pattern were analyzed following the usual protocols (*Winter, 2000*). Wing venation and pattern were recorded and genitalia of females and males were dissected. For interpretation and descriptions of genital structures we followed the procedures outlined in *Moraes & Duarte (2009)*, based on classical studies on Lepidoptera morphology (male genitalia in *Pierce, 1909*, *Sibatani et al., 1954*; *Okagaki et al., 1955*; *Klots, 1956*; *Ogata et al., 1957*; *Birket-Smith, 1974*; female genitalia in *Pierce, 1914*; *Klots, 1956*; *Mutuura, 1972*; *Galicia, Sánchez & Cordeiro, 2008*).

A total of 11 specimens belonging to the three new species here described were examined. Characters of wing venation, color pattern, and male and female genitalia were analyzed. The genitalia were illustrated with a camera lucida attached to a stereomicroscope.

Acronyms for the collections are: **MZUSP**—Museu de Zoologia da Universidade de São Paulo, São Paulo, São Paulo, Brazil; **ZUEC**—Zoological Collection of the Museu da Biodiversidade da Universidade Estadual de Campinas, Campinas, São Paulo, Brazil.

The electronic version of this article in Portable Document Format (PDF) will represent a published work according to the International Commission on Zoological Nomenclature (ICZN), and hence the new names contained in the electronic version are effectively published under that Code from the electronic edition alone. This published work and the nomenclatural acts it contains have been registered in ZooBank, the online registration system for the ICZN. The ZooBank LSIDs (Life Science Identifiers) can be resolved and the associated information viewed through any standard web browser by appending the LSID to the prefix http://zoobank.org/. The LSID for this publication is: urn:lsid: zoobank.org:pub:9450BCDC-7EB9-4CA6-BC76-04324F81ACA4. The online version of this work is archived and available from the following digital repositories: PeerJ, PubMed Central and CLOCKSS.

## RESULTS

Based on the present taxonomic sampling, the ML tree appeared divided into 10 main clades named after representative species names in each clade (Fig. 1), following *Strutzenberger et al. (2017)*. From all defined MOTUs, three of them were identified as undescribed species based on present molecular results and also on available morphological evidence, and are here described. All but one species of *Eois* were reared on species of *Piper* and the host plant species in which larvae of each MOTU were collected are shown, when available (Fig. 1). The ABGD method used for delimiting species

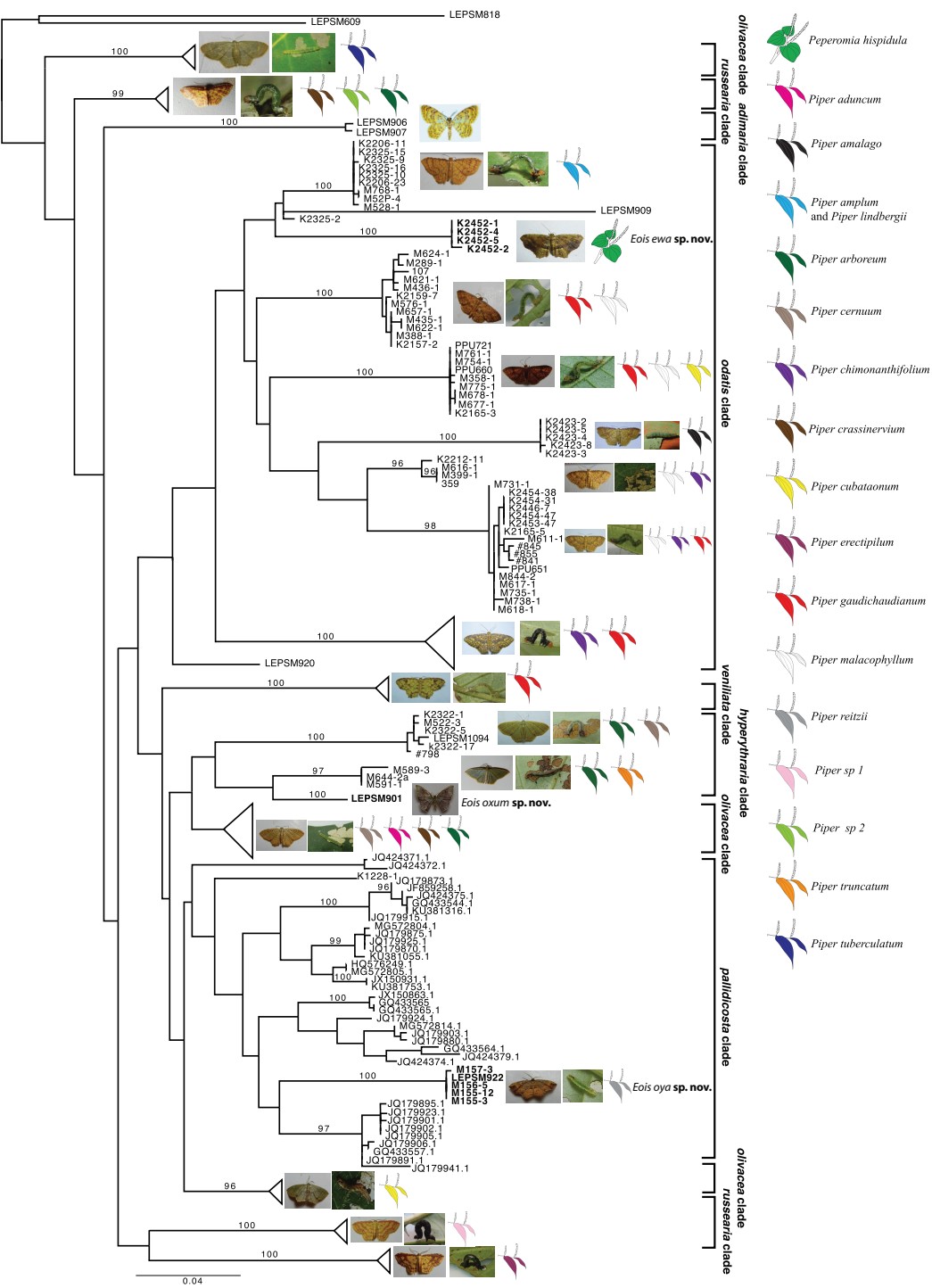

**Figure 1 Gene tree for COI-begin marker.** Colored leaves represent host plants for clades where adult representatives were obtained from reared immatures. Numbers on the node represent bootstrap stability equal or above 96%. Photographs by Simeão Moraes and Mariana Stanton.

recovered each of new species here described as a distinct molecular operational taxonomic units (MOTUs).

The first undescribed species was identified in the larger clade named "*pallidicosta* clade" (after *Eois pallidicosta* (Warren, 1907)); the high UFBoot2 support values and the long branch length (related to a genetic distance higher than 3% from all other species in the clade) suggested that this is a distinct evolutionary lineage (Fig. 1).

The second undescribed species was identified in the "*odatis* clade" (named after *Eois odatis* (Druce, 1892)). In this clade, one MOTU was recovered based on four specimens forming a well supported clade that stands out by using *Peperomia hispidula* (Sw.) A.Dietr. (Piperaceae) as larval host plant.

The third undescribed species is part of the "*hyperythraria* clade" (named after *Eois hyperythraria* (Guenée, 1858)) and is represented by a single individual collected a light source in a high montane area in Southeastern Brazil. Despite having returned to the sampling site several times no additional individuals were collected. Even though, its idiosyncratic wing pattern and the morphology of genitalia justifies the description of this new taxa based only on the holotype.

## Species Description

*Eois oya* Moraes & Montebello **sp.nov.** (Figs. 2A–2B, 3)

Diagnosis (♂ and ♀). Forewing dorsal view with a horizontal black stripe on the trunk of Cu vein, from the base of wing reaching the outer margin. Forewing and hindwing with a black dot on the discal cell closure. Aedeagus with a pointed spine close to vesica, vesica bilobed with spiniform cornuti (Fig. 3D).

Description (♂) (Figs. 2A–2B). Head: Light brown. Frons light brown, vertex light brown. Labial palp light brown. Thorax: Predominantly light brown. Prothoracic collar with iridescent gold scales. Tegulae light brown. Forewing background light brown, darker proximally; horizontal black stripe on the trunk of Cu vein, from the base of wing reaching the outer margin; one black dot on the discal cell closure; two vertical, mirrored stripes beyond discal cell closure, from costal margin to inner margin; submarginal band as a faint stripe; marginal band following the outer margin contour; underside with the same dorsal pattern. Hindwing with the same forewing pattern, except being lighter proximally and without the submarginal stripe. Abdomen. Dorsally brown; ventrally light brown with two lateral dark brown stripes. Genitalia (Figs. 3A–3D): Tegumen triangular in dorsal view, with the anterior margin round. Uncus absent. Valva entitre, sub-elliptical; sacculus developed, consisting of an anterior projection with rounded apex. Labides absent. Fultura inferior or juxta sclerotized, shaped like an inverted "U". Saccus with a short anterior projection. Subscaphium smooth. Aedeagus rectilinear with a spine near the vesica; ejaculatory bulb rounded, foramen lateral; vesica bilobed, lobes with dense sclerotized spiniform cornuti.

Description (♀) Head: Same as in the male except for the antenna less pectinated. Thorax: Same as in the male. Abdomen: Same as in the male. Genitalia (Fig. 3E): Seventh sternite smooth; ostium membranous not fused with the seventh sternite; antrum short and membranous, except for a sclerotized bracket-shaped support close to corpus bursae;

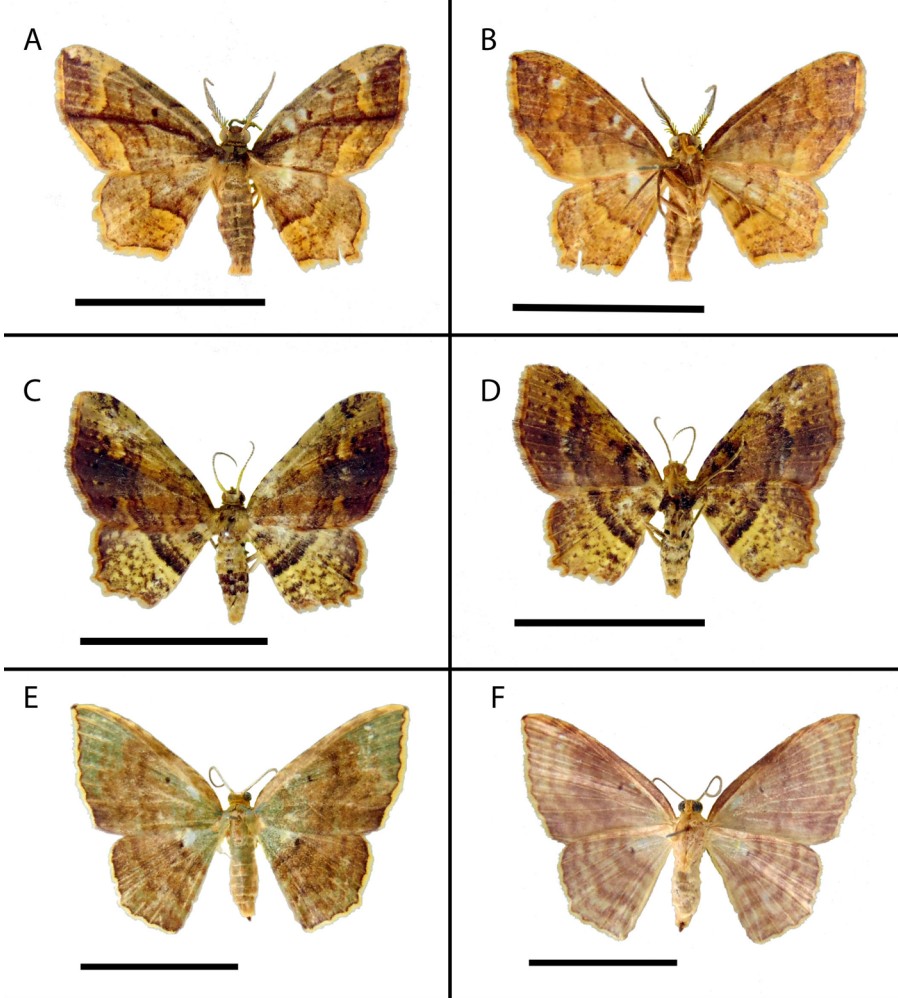

**Figure 2 Habitus of *Eois* holotype specimens.** A–B. Male, holotype of *Eois* oya **sp.nov.** A. Dorsal view. B. Ventral view. C–D. Male, holotype of *Eois* ewa **sp.nov.** C., Dorsal view. D. Ventral view. E–F. Female, holotype of *Eois* oxum **sp.nov.** Scale bar 1 cm. Photographs by Simeão Moraes.

ductus bursae short and membranous; corpus bursae extending beyond the seventh sternite; signa consisting of several microspicles and a falciform spine. Bulla seminallis arising from a ventral pouch on the posterior portion of corpus bursae. Lamella antevaginalis and postvaginalis absent.

Etymology. The specific epithet, *oya* is the Brazilian name for the female orisha who commands the winds, lightning and storms. In the native culture of the Yoruba people, orishas represent spirits sent for the guidance of all creation and of humanity.

The Portuguese spelling comes from the Yoruba "Ọya" which means "she tore". She is the patron of the Niger River, known to the Yoruba as the Odo-Ọya. The specific epithet is a tribute to women and to Brazilian black culture. A noun in apposition.

Distribution. The few records for this species are from medium and low altitudes (800 m to 1,200 m a.s.l.) in the Serra do Mar and Serra da Mantiqueira mountain chains, in a

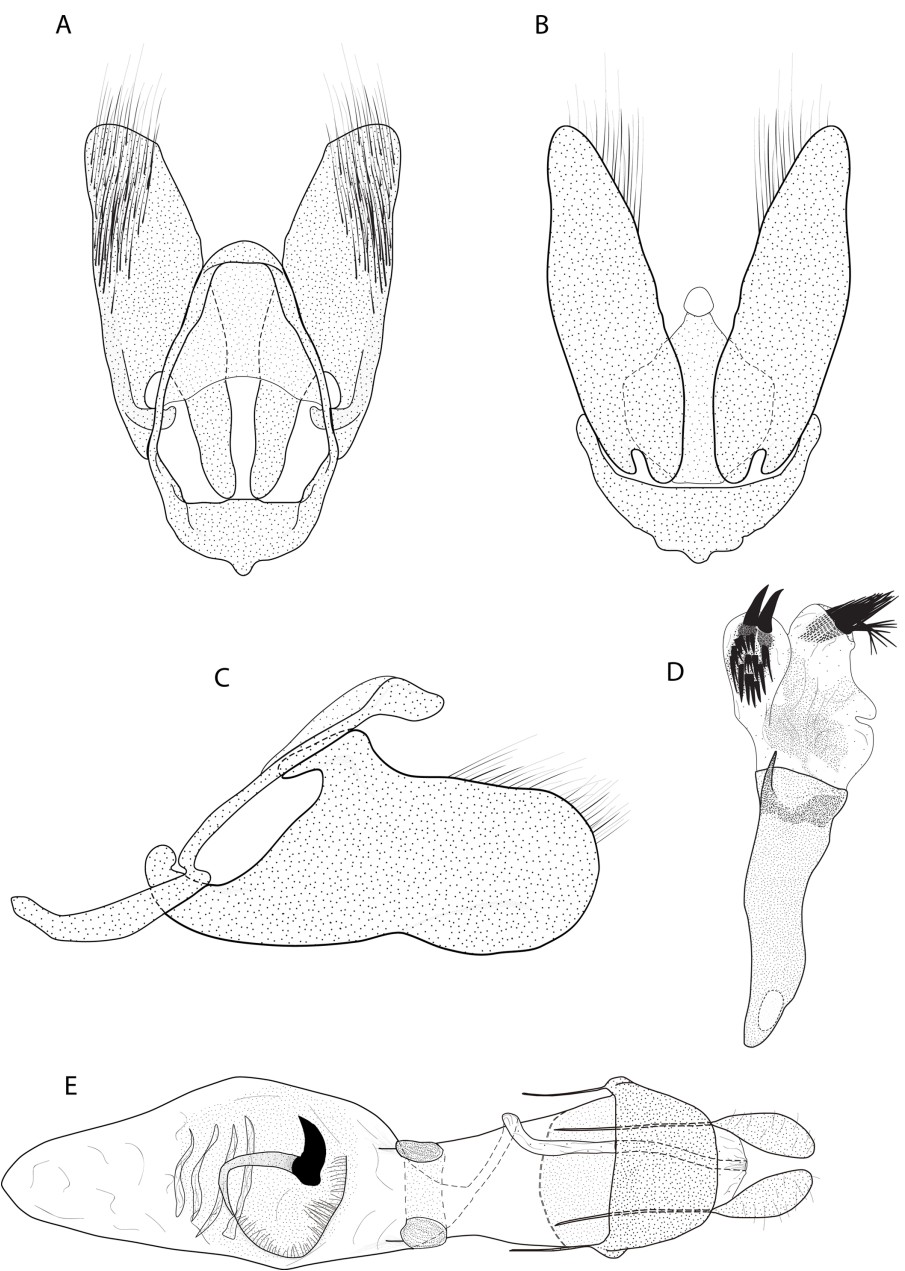

**Figure 3 Male and female genitalia of *Eois oya* sp.nov. paratype.** A. Male genitalia, genital capsule, dorsal view. B. Male genitalia, genital capsule, ventral view C. Male genitalia, genital capsule, lateral view. D. Male genitalia, aedeagus, lateral view. E. Female genitalia, dorsal view. Line drawings by Simeão Moraes and Ygor Montebello                 

narrow region of the Atlantic Forest near the border between the states of São Paulo and Rio de Janeiro.

Remarks: Some adults were obtained from immature stages hand-collected on *Piper reitzii* plants at the Parque Nacional do Itatiaia, in the state of Rio de Janeiro and reared to adults in laboratory (see "Methods" section).

Type series. HOLOTYPE ♂, ex larva: BRAZIL: Rio de Janeiro: Itatiaia:, Parque Nacional do Itatiaia 22° 27′ 01.5″ S 44° 37′ 14.0″ W, 1,174 m asl, 03-VIII-2016, Simeão M., Tamara A. & Mariana S leg. Deposited in the Zoological Collection of the Museu da Biodiversidade da Universidade Estadual de Campinas (ZUEC), Campinas, São Paulo, Brazil.

PARATYPES (all from Brazil): *Rio de Janeiro*: **Itatiaia**, 1♂ and 2 ♀, ex larva, Parque Nacional do Itatiaia, 22° 27′ 01.5″ S 44° 37′ 14.0″ W, 1,174 m asl, 03-VIII-2016, Simeão M., Tamara A. & Mariana S. leg. (ZUEC). *São Paulo*: **Salesópolis**, 1♂ and 1♀, Estação Biológica de Boraceia, 23° 39′S 45° 54′W, 850 m asl, 28-X/ 01-XI-2016, Simeão M., Tamara A. & André T leg. (MZUSP).

*Eois ewa* Moraes & Stanton **sp.nov.** (Figs. 2C–2D, 4)

Diagnosis (♂ and ♀). Forewing dorsal view with dark brown maculae on the outer margin. Forewings and hindwings with two vertical mirrored bands on discal cell closure, continuous with dorsally dark brown abdominal segments A5 and A6 (Fig. 2C).

Description (♂) (Figs. 2C–2D). Head: Brown. Frons brown, vertex brown. Labial palp light brown. Thorax: Predominantly light brown. Prothoracic collar with iridescent gold scales. Tegulae light brown. Forewing background rusty brown; two sinuous black stripes on the wing base, from de trunk of R vein, reaching the inner margin; two vertical mirrored bands on discal cell closure, from the trunk of R4+R5 to inner margin; dark brown maculae on the outer margin, merged with the discal bands in the region of discal cell closure; underside with the same dorsal pattern. Hindwing with the same forewing pattern, except with the light brown background and the outer maculae replaced by three bands composed of brown spots. Abdomen: Dorsally brown; dark brown central macula on abdominal tergites A1-A4, abdominal tergites A5-A6 dark brown; ventrally light brown with patches of dark brown scales on the sternite A2-A8 margin. Genitalia (Figs. 4A–4D): Tegumen triangular in dorsal view, with the anterior margin round. Uncus absent. Valva trapezoidal; sacculus developed, consisting of an anterior projection with rounded apex. Labides absent. Transtila sclerotized, squared. Fultura inferior or juxta sclerotized, shaped like a "U". Saccus with a short anterior projection. Aedeagus rectilinear and smooth; ejaculatory bulb rounded, foramen lateral; vesica bilobed, lobes with two patches of sclerotized spiniform cornuti.

Description (♀). Head: Same as in the male except for the antenna less pectinated. Thorax: Same as in the male. Abdomen: Same as in the male. Genitalia (Fig. 4E): Seventh sternite smooth; ostium membranous not fused with the seventh sternite; antrum short and membranous, except for a sclerotized ring close to corpus bursae; ductus bursae short and membranous; corpus bursae with multiseriated signa, signa consisting of several microspicles displaced at the anterior portion of corpus bursae. Accessory bag smooth. Lamella antevaginalis and postvaginalis absent.

Etymology. The specific epithet, *ewa* comes from Yoruba "Yewá". In Brazil, ewa is name for the female orisha and river deity from the Yewá river, located in the ancient Egbado tribe (present-day city of Yewa). In the native culture of the Yoruba people, orishas represent spirits sent for the guidance of all creation and of humanity.

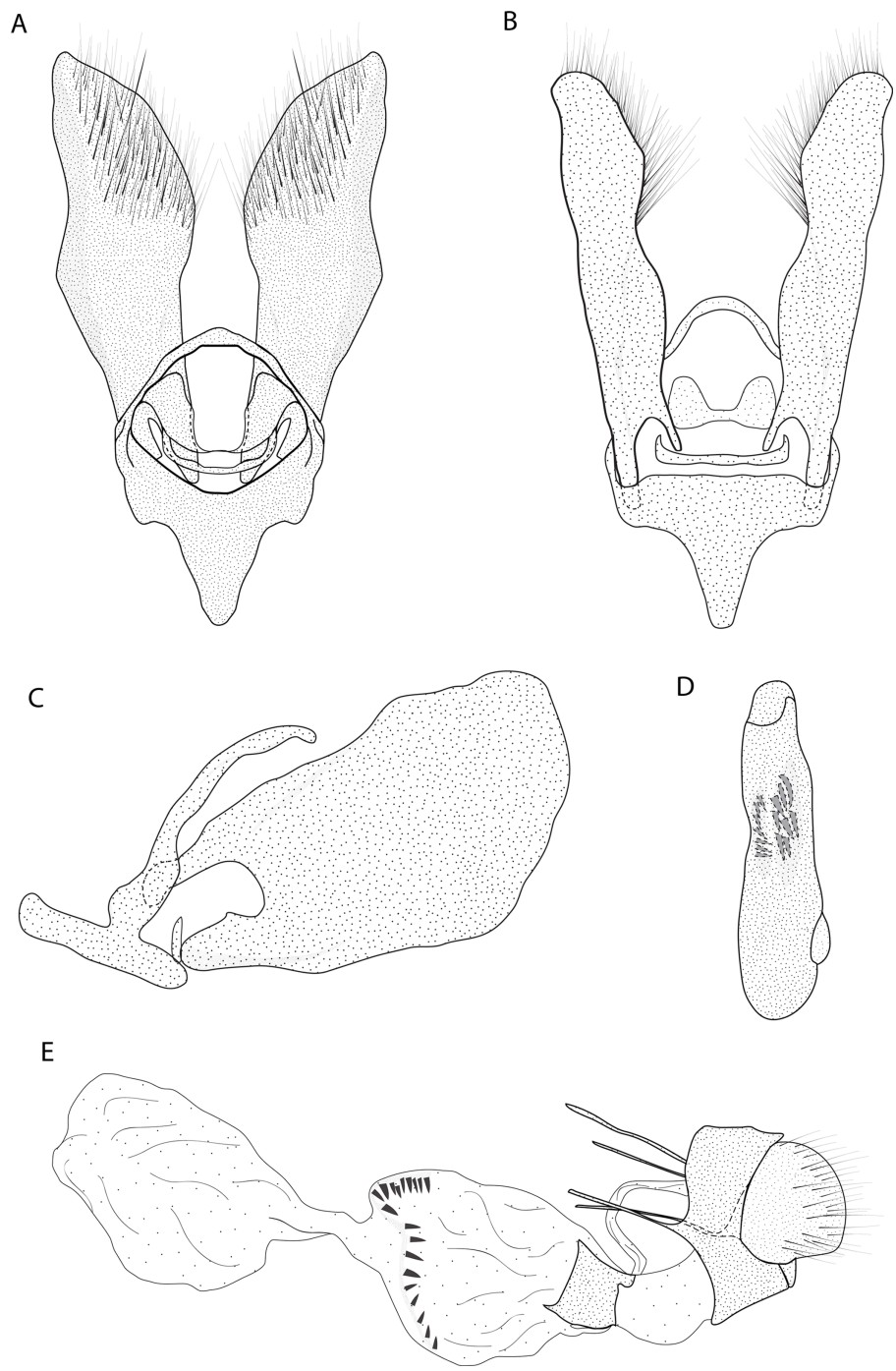

**Figure 4  Male and female genitalia of *Eois ewa* sp.nov. paratype.** A. Male genitalia, genital capsule, dorsal view. B. Male genitalia, genital capsule, ventral view C. Male genitalia, genital capsule, lateral view. D. Male genitalia, aedeagus, lateral view. E. Female genitalia, lateral view. Line drawings by Simeão Moraes.                                           

Ewa represents the gift of divination and intuition. She also represents the mutations, transformations and the perception of what is beautiful and what is ugly. The specific epithet is a tribute to women and to Brazilian black culture. A noun in apposition.

Distribution. The four individuals known were obtained from reared larvae collected on the host plant *Peperomia hispidula* in humid montane forests (altitude of 1,100 m a.s.l.) at the Itatiaia National Park, in the state of Rio de Janeiro.

Type series. HOLOTYPE ♂, ex larva: Rio de Janeiro: Itatiaia: Parque Nacional do Itatiaia, 22° 25′ 37.8″ S 44° 37′ 07.0″ W, 1,100 m asl, 30-VI-2017, Lydia Y., Mariana S. & Simeão M. leg. Deposited in the Zoological collection of the Museu da Biodiversidade da Universidade Estadual de Campinas (ZUEC), Campinas, São Paulo, Brazil.

PARATYPES (all from Rio de Janeiro, Brazil): **Itatiaia**: 2♂ and 4♀, ex *larva*, Parque Nacional do Itatiaia, 22° 25′ 37.8″ S 44° 37′ 07.0″ W, 1,100 m asl, 30-VI-2017, Lydia Y., Mariana S. & Simeão M. leg. (ZUEC); 1♀, ex larva: 30-VI-2017, Lydia Y., Mariana S. & Simeão M. leg. (MZUSP).

*Eois oxum* Moraes & Freitas **sp.nov.** (Figs. 2E–2F, 5)

Diagnosis (♀). Forewing costal margin brown, olive green at the wing base and apex, in dorsal view. Hindwing upperside olive green at the wing base. Forewing and hindwing with a black dot on the center of discal cell. Abdominal segments A1 and A2 olive green dorsally (Fig. 2E).

Description (♀) (Figs. 2E–2F). Head: Brown. Frons brown, vertex brown. Labial palp light brown. Thorax: Predominantly olive green. Prothorax brown and prothoracic collar olive green. Tegulae olive green. Forewing background olive green, costal margin brown; a black dot in the center of discal cell; two vertical mirrored dark brown stripes in the discal cell closure, from costal margin to inner margin; two post discal bands, faint from costal margin to $R_5$, dark brown from $R_5$ to inner margin; dark brown maculae on the tornus region, merged with the post discal bands on the region close to tornus; outer margin delineated by rusty brown scales followed by a fringe of yellow scales; underside with the same dorsal pattern but with light brown background and rusty brown bands. Hindwing with the same forewing pattern, except for the darker apex and the delineated post discal bands. Abdomen: Dorsally olive green on segments A1-A2, beige on segments A3-A7, ventrally beige. Genitalia (Fig. 5): Seventh sternite smooth; ostium sclerotized not fused with the seventh sternite; antrum short and membranous, except for a sclerotized ring close to corpus bursae; ductus bursae short and membranous; corpus bursae extending beyond the seventh sternite, signa consisting of several microspicles and a falciform spine. Bulla seminallis arising from a ventral pouch on the posterior portion of corpus bursae Lamella antevaginalis absent. Lamella postvaginalis sclerotized, square shaped.

Etymology. The specific epithet *oxum* comes from Yoruba "Oṣun". It is the Brazilian name of the female orisha and river deity who reigns over fresh waters. In the native culture of the Yoruba people, orishas represent spirits sent for the guidance of all creation and of humanity.

Oxum is considered the lady of beauty, fertility, money and sensitivity. Its name derives from the Oṣun River, which flows in Yorubaland, the Nigerian region of Ìjẹṣà. The specific epithet is a tribute to women and to Brazilian black culture. A noun in apposition.

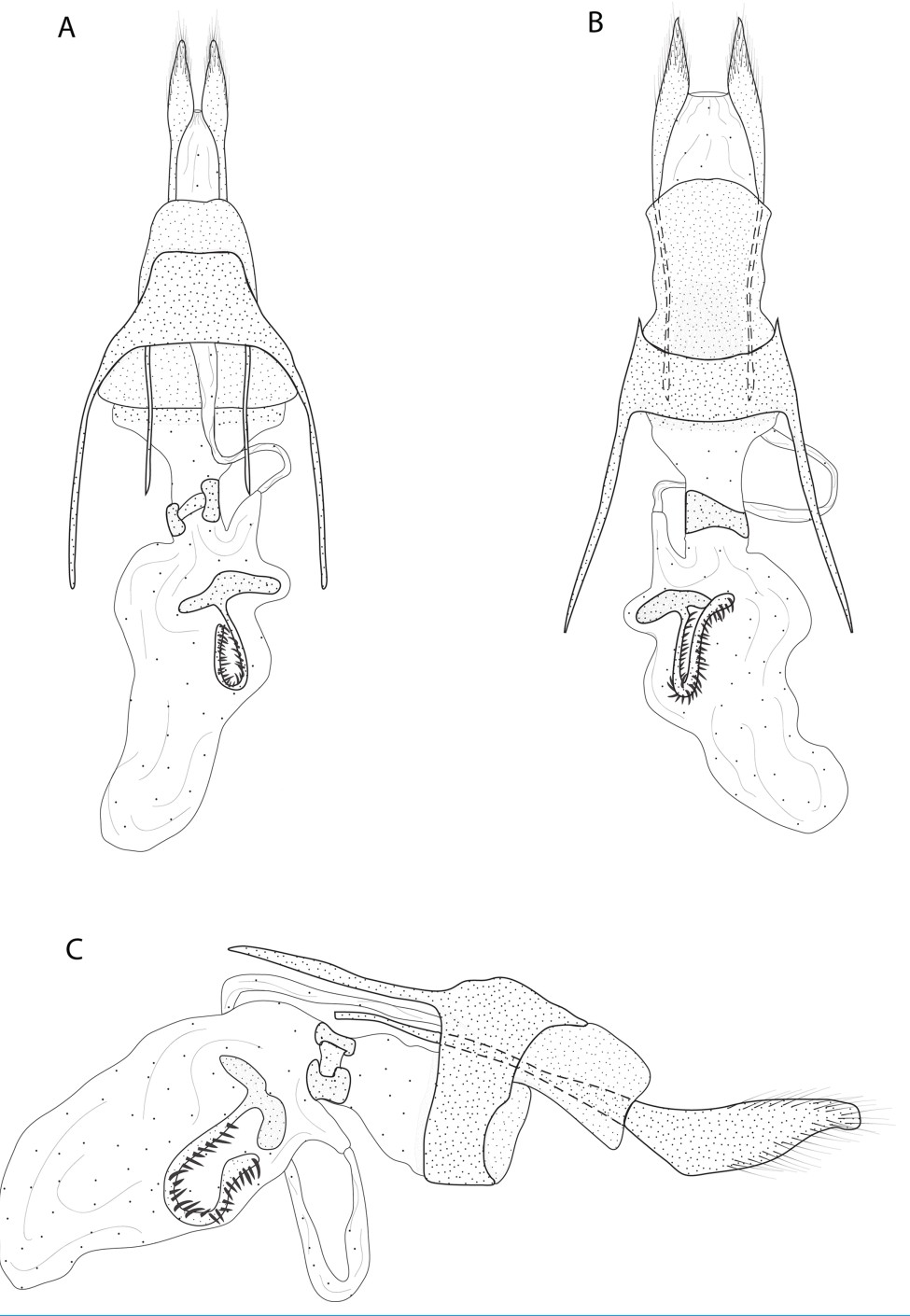

**Figure 5 Female genitalia of *Eois oxum* sp.nov. holotype.** A. Genitalia, dorsal view. B. Genitalia, ventral view C. Genitalia, lateral view. Line drawings by Simeão Moraes

Distribution. The single record came from a site of montane rainforest in a region with elevations ranging from 800 m to 1,000 m a.s.l. in the Serra do Mar mountain chain, in São Paulo State.

Remarks: *Eois oxum* is represented by a singleton collected on a lightrap. Despite having returned to the sampling site several times no additional individuals were collected. Regardless of having just one specimen available, the idiosyncratic wing pattern and the morphology of genilalia justifies the description of this new taxon based only on the holotype.

Type series. HOLOTYPE ♀: BRAZIL: São Paulo: Salesópolis, Estação Biológica de Boraceia, 23° 39′ S 45° 54′ W, 850 m asl, 28-X/ 01-XI-2016, Simeão M., Tamara A. & André T leg. Deposited in the Zoology Collection of the Museu da Biodiversidade da Universidade Estadual de Campinas (ZUEC), Campinas, São Paulo, Brazil.

## DISCUSSION

Although the diversity of *Eois* has been highlighted in previous studies based on molecular evidence (*Strutzenberger et al., 2011*; *Wilson et al., 2012*; *Moraes et al., 2021*), the lack of researchers working on this diversity and providing stability for the names through formal description of new taxa precluded a comprehensive taxonomic treatment so that the proper description of this huge diversity remains a taxonomic challenge.

In the present study, we provided a preliminary taxonomic assessment for a small clade of *Eois*; a broader taxonomic sampling and additional evidence (molecular and morphological) will be needed to deal with larges clades, such as those of *E. olivacea* and *E. tegularia*. In short, although the present study is a small contribution for a clade that is clearly composed of a large number of species, including several complexes of cryptic species, we hope that it contributes to a better understanding of the genus *Eois*. Moreover, we believe that forthcoming studies in this group can take advantage of several sources of evidence, including morphology, molecular data and host plant use among other not yet properly explored (e.g., data from immature stages). In this context of integrative taxonomy, the distinct evolutionary lineages (i.e., distinct species) can be better recognized and defined, unveiling the real biodiversity of this large genus of moths.

## ACKNOWLEDGEMENTS

We are grateful to André Rangel Nascimento and Tara Joy Massad for providing some of the *Eois* samples used here, to Thamara Zacca for critical reading of the manuscript and to Eric J. Tepe for providing identification for some Piper species used in this study. This study is part of the project Dimensions US-Biota São Paulo: "Chemically mediated multi-trophic interaction diversity across tropical gradients" (Fapesp 2014/50316-7).

### Funding

This work was supported by the Fundação de Amparo à Pesquisa do Estado de São Paulo (FAPESP) with the scholarship awarded by SSM (grants FAPESP 2015/17047–5, FAPESP 2016/20196–5), YZM (2019/02727-1) and MAS (FAPESP 2015/26823-9). AVLF thanks the Brazilian Research Council—CNPq (grant 303834/2015-3), the FAPESP (grants no. 2011/50225-3, 2012/50260-6, 2013/50297-0) and The United States Agency for

International Development—USAID/the U.S. National Academy of Sciences (NAS), under the PEER program (Sponsor Grant Award Number: AID-OAA-A-11-00012). The funders had no role in study design, data collection and analysis, decision to publish, or preparation of the manuscript.

## Grant Disclosures

The following grant information was disclosed by the authors:
Fundação de Amparo à Pesquisa do Estado de São Paulo (FAPESP): FAPESP 2015/17047–5, FAPESP 2016/20196–5.
YZM: 2019/02727-1.
MAS: FAPESP 2015/26823-9.
CNPq: 303834/2015-3.
FAPESP: 2011/50225-3, 2012/50260-6, 2013/50297-0.
U.S. National Academy of Sciences (NAS): AID-OAA-A-11-00012.

## Competing Interests

The authors declare that they have no competing interests.

## Author Contributions

- Simeão S. Moraes conceived and designed the experiments, performed the experiments, analyzed the data, prepared figures and/or tables, and approved the final draft.
- Ygor Z. Montebello conceived and designed the experiments, performed the experiments, analyzed the data, prepared figures and/or tables, and approved the final draft.
- Mariana A. Stanton conceived and designed the experiments, performed the experiments, analyzed the data, prepared figures and/or tables, and approved the final draft.
- Lydia Fumiko Yamaguchi performed the experiments, authored or reviewed drafts of the paper, and approved the final draft.
- Massuo J. Kato performed the experiments, authored or reviewed drafts of the paper, and approved the final draft.
- André V.L. Freitas conceived and designed the experiments, analyzed the data, authored or reviewed drafts of the paper, and approved the final draft.

## Field Study Permissions

The following information was supplied relating to field study approvals (i.e., approving body and any reference numbers):
   Field permits were granted by ICMBio (Instituto Chico Mendes para Biodiversidade) (permit number 10362-1, permit number 15780-10 and permit number 22205-6).

## DNA Deposition

The following information was supplied regarding the deposition of DNA sequences:
   The new generated sequences are available at GenBank and in the Supplementary File. The accessions from sequences retrieved from Genbank are also available in Table 1:

GQ433544.1, GQ433557.1, GQ433564.1, GQ433565, GQ433565.1, HQ576249.1, JF859258.1, JQ179870.1, JQ179873.1, JQ179875.1, JQ179880.1, JQ179891.1, JQ179895.1, JQ179901.1, JQ179902.1, JQ179903.1, JQ179905.1, JQ179906.1, JQ179915.1, JQ179923.1, JQ179924.1, JQ179941.1, JQ424371.1, JQ424372.1, JQ424374.1, JQ424375.1, JQ424379.1, JX150863.1, JX150931.1, KU381055.1, KU381316.1, KU381753.1, MG572804.1, MG572805.1, MG572814.1.

## Data Availability

The raw data are available in the Supplementary File.

The holotypes and paratypes of new described species are deposited at Zoological Collection of the Museu da Diversidade Biológica da Universidade Estadual de Campinas (ZUEC), Campinas, São Paulo, Brazil (accession numbers LEPSM901, K2452-5, K2452-2, M155-12, M155-3, LEPSM922) and at Museu de Zoologia da Universidade de São Paulo (MZSP), São Paulo, São Paulo, Brazil (accession numbers K2452-1, K2452-4, M157-3, M156-5).

## New Species Registration

The following information was supplied regarding the registration of a newly described species:

Publication LSID: urn:lsid:zoobank.org:pub:9450BCDC-7EB9-4CA6-BC76-04324F81ACA4

Eois ewa Moraes & Stanton LSID: urn:lsid:zoobank.org:act:A0C8118B-29C7-41E1-8434-5FFDEBFB3064,

Eois oxum Moraes & Freitas LSID: urn:lsid:zoobank.org:act:CB95F8D8-E007-42D5-8C76-3A3FEE15D967.

Eois oya Moraes & Montebello LSID: urn:lsid:zoobank.org:act:C17CD8C7-D933-4C9E-8307-F74CC05D6CA1.

## Supplemental Information

Supplemental information for this article can be found online at http://dx.doi.org/10.7717/peerj.11304#supplemental-information.

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
