# Peer review of "Description of three new species of Geometridae (Lepidoptera) using species delimitation in an integrative taxonomy approach for a cryptic species complex"

_PeerJ, doi:10.7717/peerj.11304_

## Round 0.1 · original submission · Minor Revisions

Dear Drs. Moraes and Freitas,

Congrats! After three independent evaluations, the reviewers considered your manuscript is publishable and meets PeerJ's standard for publication. Nonetheless, they believe you need to perform minor revisions in the document, to make it ready for publication in PeerJ

Best regards,
Daniel Silva

Reviewer 1 ·

Basic reporting

line 106: Moraes et al. 2016 is missing.

line 217: the use of term “juxta”: according to Niculescu (1972, 1988), the term “juxta” should not be used, once “fultura inferior” was earlier proposed by Petersen (1904), and were replaced for juxta by Pierce (1909, 1914).

Experimental design

no comment.

Validity of the findings

no comment.

Additional comments

Dear authors:

The manuscript entitled ‘Description of three new species of Geometridae using species delimitation in an integrative taxonomy approach for a cryptic species complex’ is an important contribution towards the knowledge of the systematics of Neotropical Geometridae.

I made some minor comments and suggestions that should not compromise the acceptance of the manuscript.

Reviewer 2 ·

Basic reporting

This manuscript is focused on a very diverse genus of moths from the Neotropics. The authors take up an integrative taxonomy approach in species delineation by combining DNA barcode sequences with ecological information on the larval host plants, as well as a careful examination of relevant morphological characters. The text is clear and very well written with some minor problems that I tried to correct in the Word version of the manuscript (uploaded separately). The authors provided a good background and have cited the relevant works appropriately. The article is overall well structured and the raw data are shared. The illustrations of the genital structures are beautiful as well as very informative and the photographs of the adults are done at a high level, clearly showing all of the relevant features. The tree figure needs to be improved and I provide additional comments on that below.

Experimental design

The research question is clearly stated and defined. It is highly relevant given the accelerated extinctions that are ongoing all around the planet and as has been shown recently, insects are not spared. It is important that alpha taxonomy is carried out and that we document as much as possible of our planet’s biodiversity before it disappears. We can’t do much to protect this diversity if we don’t know that it exists! The methods are described with enough details to replicated the study and the methodology is sound.

Validity of the findings

The authors make a good case that they have discovered new species of the genus Eois in Brazil and the description of these three new species is fully justified. We have evidence from several different sets of characters: DNA barcode, external morphology of the adults and genitalic structures, and host plant specialization. Perhaps the number of reared individuals is quite low to allow for a full assessment of how useful the host plant data is at this level, but it does give additional data that are useful.

Additional comments

Great job overall, but as I stated above, the tree figure needs additional work. Firs, the font is much too small to be readable. It should be 2-3 times larger at the minimum! I understand that most of us these days view papers on our computers, where we can zoom in as necessary, but here, when one zooms in enough to read the text, one can only see a small part of the tree, and given how much empty space there is in this figure overall, you can do a much better job. You can also double or triple the size of the pictures of the moths and larvae, and perhaps somewhat increase the size of the host plant symbols, although they can even remain the same size.

Also, it’s unclear to me whether each of the DNA barcoding clusters represents an already described species or more still unnamed species. For example, odatis clade seem to include several species. It would be helpful if you could label all of these clusters with names of at least the described species. I assume that one of these lineages is Eois odatis.

And the tree figure would overall be easier to read if it were displayed in increasing or decreasing order.

Annotated reviews are not available for download in order to protect the identity of reviewers who chose to remain anonymous.

Reviewer 3 ·

Basic reporting

The manuscript is very well writen; clear and precise.In the introduction of the work, the authors make clear the structure and purpose of the study.
The structure of the article follwes the format of the PeerJ Journal. Table and figures high-quality. Raw data shared.

Experimental design

The manuscript reveals a original and relevant study since contributes to understanding about a complex of cryptic species of neotropical moths.
Moreover, using morphology, host plant use and species delimitation tools proved to be very appropriate for the study.

Validity of the findings

No comment.

Additional comments

Studying neotropical moths is challenge given its enormous diversity. In their study you describe three new species (Eois genus) and provide a relevant dataset. The scientific value of the information is excellent.

---

## Round 0.2 · accepted · Accept

Dear authors!

Congratulations! All three reviewers indicated your manuscript should be accepted for publication! Well done!

Reviewer 2 ·

Basic reporting

Looks good.

Experimental design

Looks good.

Validity of the findings

All good.

Additional comments

Looks great! I like how the changes in this version were implemented based on the review comments.

Reviewer 3 ·

Basic reporting

As a results of the revision (minor revision), I consider the manuscript has been greatly improved.

Experimental design

No comment.

Validity of the findings

No comment.

Additional comments

I consider your manuscript as a beautiful incentive for future studies of Neotropical species of Macroheterocera. Congratulations!